# Glycans unique to the relapse-prone subset within triple-negative breast cancer as revealed by lectin array-based analysis of surgical specimens

Madoka Sakata-Matsuzawa[1], Kaori Denda-Nagai[2]*, Haruhiko Fujihira[2,3], Miki Noji[2], Katrin Beate Ishii-Schrade[2], Atsushi Matsuda[4], Atsushi Kuno[5], Misato Okazaki[1], Katsuya Nakai[1], Yoshiya Horimoto[1], Mitsue Saito[1], Tatsuro Irimura[2]*

1 Department of Breast Oncology, Juntendo University School of Medicine, Tokyo, Japan, 2 Division of Glycobiologics, Intractable Disease Research Center, Juntendo University Graduate School of Medicine, Tokyo, Japan, 3 Glycometabolic Biochemistry Laboratory, Cluster for Pioneering Research, RIKEN, Saitama, Japan, 4 Department of Biochemistry, Keio University School of Medicine, Tokyo, Japan, 5 Molecular & Cellular Glycoproteomics Research Group, Cellular and Molecular Biotechnology Research Institute, National Institute of Advanced Industrial Science & Technology, Tsukuba, Ibaraki, Japan

* t-irimura@juntendo.ac.jp (TI); k-denda@juntendo.ac.jp (KDN)

**Data Availability Statement:** All relevant data are within the manuscript and its Supporting information files.

## Abstract

### Introduction

Molecular and cellular characteristics of the relapse-prone subset within triple-negative breast cancer (TNBC) remain unclear. Aberrant glycosylation is involved in the malignant behavior of cancer cells. In the present study, we aimed to reveal glycan profiles unique to relapsed TNBC patients.

### Methods

Thirty TNBC patients who did not undergo neoadjuvant chemotherapy but postoperative standard adjuvant therapy from 2009 through 2016 at Juntendo Hospital were investigated. TNBC cells were resected from primary breast cancer sections of formalin-fixed surgical specimens using laser-assisted microdissection. The binding intensities of the extracted glycoproteins to 45 lectins were quantified using lectin microarray and compared between relapsed and non-relapsed patients. Immunohistochemical staining with TJA-II lectin in specimen sections was performed.

### Results

Five patients relapsed during the follow-up (range 37–123 months). Lectin microarray analysis revealed that 7 out of 45 lectins showed significant differences in binding intensity between the relapsed and the non-relapsed group. TJA-II, ACA, WFA, and BPL showed stronger binding in the relapsed group. PNGase F treatment of TNBC cell lysates suggested that TJA-II and ACA bind O-glycans. TJA-II staining of tissue sections revealed strong binding to cell surface membranes and to the cytoplasm of TNBC cells, but not to other types of

**Funding:** This work was supported as part of the Japan Agency for Medical Research and Development (AMED) Project by the Ministry of Economy, Trade and Industry (METI) (JP19ae0101026 to TI). This work was also supported in part by Japan Society for the Promotion of Science Core-to-Core program entitled International Core Research Center for Phototheranostics (JPJSCCA20170007 to TI). The funders had no role in study design, data collection and analysis, decision to publish, or preparation of the manuscript.

**Competing interests:** The authors have declared that no competing interests exist.

cells. Significantly more TNBC cells were stained in tissue sections from relapsed than non-relapsed patients.

## Conclusions

TNBC cells from relapsed patients showed a unique lectin reactivity, with higher levels of TJA-II (also WFA and BPL) binding than in non-relapsed patients. The results are potentially useful to develop new prognostic and therapeutic tools.

## Introduction

Breast cancer is one of the most common malignant tumors among women. Triple-negative breast cancer (TNBC) is a subtype of breast cancer that accounts for about 15–20% of all breast cancers and is defined by tumors lacking estrogen receptor expression, progesterone receptor expression, and human epidermal growth factor receptor-2 (HER2) overexpression/amplification [1, 2]. As an adjuvant therapy, cytotoxic chemotherapy is the only available treatment option for TNBC patients because they do not respond to hormone or anti-HER2 treatment. However, a subset of TNBC patients resists chemotherapy, resulting in shorter relapse-free survival and poorer prognosis compared to the rest of TNBC patients or other breast cancer subtypes [3, 4]. Recently, progress in the identification of potential therapeutic target molecules for TNBC has been made [5, 6], including one study showing that the PARP (polyadenosine diphosphate-ribose polymerase) inhibitor, talazoparib, exhibited greater antitumor activity than standard chemotherapy among patients with advanced breast cancer and a germline *BRCA1/2* mutation [7]. Immune check point inhibitors represent another recent progress in TNBC treatment, however, the indication is limited to cases with high levels of PD-L1 (programmed cell death ligand 1) or high microsatellite instability [8]. Nevertheless, studies aimed at exploring the molecular properties associated with the relapse-prone subset of TNBC remain scarce.

Glycosylation is one of the major co- and post-translational modifications of proteins, which are catalyzed essentially by glycosyltransferases. Changes in protein glycosylation modify the functions of proteins involved in the malignant behavior of cancer cells, including cell-cell interactions leading to cancer cell dissemination and local invasion, regulation of the immune response, and regulation of growth in a variety of organ microenvironments [9–14]. Regarding the implications of glycosylation in drug sensitivity, altered glycosylation of ATP binding cassette transporters has been reported to be associated with drug-resistant tumor cells [15, 16]. Altered glycosylation during breast cancer progression and metastasis is also observed [17].

Previous studies which explored the relationship between the malignant behavior of TNBC cells and glycosylation came to some interesting conclusions. For example, a study by Zhou and co-workers reported that bindings of *Ricinus communis* agglutinin I (RCA-I) to TNBC cell lines correlated with their metastatic capacity, and RCA-I inhibited the adhesion, migration, and invasion of TNBC cells with highly metastatic capacity [18]. It is notable that most of the previous studies on this subject were conducted *in vitro* with breast cancer cell lines.

In the present study, we used laser-assisted microdissection to resect TNBC cells from 30 surgical specimens and compared the binding intensities of extracted glycoproteins to 45 lectins between relapsed and non-relapsed TNBC specimens by the use of lectin microarrays. Our data provide information on the unique lectin reactivity of TNBC cells from relapse-

prone patients, which we believe, will eventually lead to the identification of a unique glyco-proteome, which in turn will be useful to develop diagnostic and therapeutic antibodies to specifically identify and treat this patient population.

## Materials and methods

### Patient selection

In the present study, 66 TNBC patients who underwent curative breast surgery during the period from August 1, 2009 through March 31, 2016 at Juntendo University Hospital were investigated. From this initial pool of candidates, which comprised Stage I to III patients, we further selected TNBC patients according to the following criteria. First, to ensure that tumor characteristics were not affected by presurgical treatments, patients who received preoperative chemotherapy were excluded. Second, to ensure that the relapse was not influenced by differences in the treatment regimen, only patients to whom standard adjuvant therapy for TNBC was administered after surgery were included. Among TNBC patients who met these two criteria, 30 patients whose surgical specimens were available were included into the data analysis (Fig 1). In the present study, estrogen receptor and progesterone receptor status were defined as negative when less than 1% of the nuclei of cancer cells were stained, using immunohistochemistry (IHC). HER2 status was defined as negative when the IHC score was 0 or 1+. In addition, patients who had a HER2 IHC score of 2+, but whose HER2/*neu* gene amplification was not confirmed by fluorescence in situ hybridization, were also defined as HER2-negative [19]. All patients included in the present study had received a standard postoperative chemotherapy treatment. Twenty-seven patients had received a FEC regimen (fluorouracil 500 mg/$m^2$, epirubicin 75–100 mg/$m^2$, and cyclophosphamide 500 mg/$m^2$ administered every 21 days for 4–6 cycles). Seven patients with lymph node metastasis and 3 patients considered at high risk of relapse had received a taxane regimen (docetaxel 75 mg/$m^2$ every 21 days for 4 cycles or paclitaxel 80 mg/$m^2$ once per week for 12 weeks) in addition to FEC regimen. Furthermore, 3 patients (elderly patients and patients with cardiac dysfunction) who could not receive the

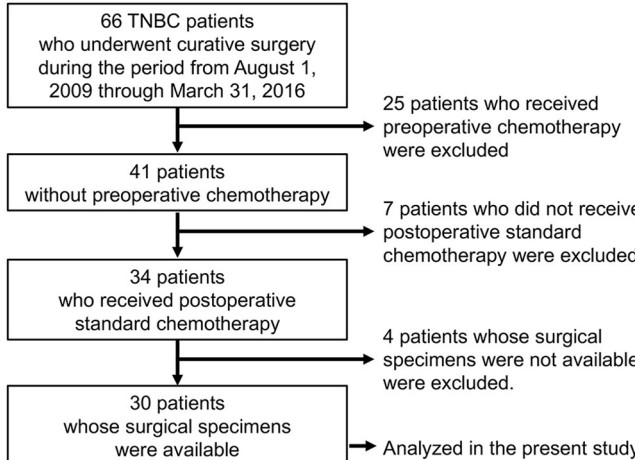

**Fig 1. Patient selection flow chart.** TNBC patients who did not undergo neoadjuvant chemotherapy but postoperative standard adjuvant therapy from 2009 through 2016 at Juntendo Hospital were investigated in the present study. TNBC; triple-negative breast cancer.

FEC regimen had received a TC regimen (docetaxel 75 mg/m$^2$ and cyclophosphamide 600 mg/m$^2$ every 21 days for 4 cycles).

## Laser-assisted microdissection of TNBC tissue and protein extraction

TNBC tissue samples were obtained from primary breast surgical specimens. Laser microdissection and protein extraction were performed as previously described [20–22].

Briefly, Hematoxylin staining was performed after deparaffinizing the foil slides (Leica Microsystems, Wetzlar, Germany) of 4 μm-thick surgical specimens. Using LMD7000 (Leica Microsystems), the area containing tumor cells was outlined according to the Hematoxylin Eosin staining, and tumor cells were removed by laser and collected into a tube containing 200 μL of 10 mM citrate buffer (LSI Medience, Tokyo, Japan). An area of 2.5 mm$^2$ was processed for analysis. The solution containing dissected tumor cells was incubated for 1 h at 95˚C. After cooling on ice, 4 μL of 50% (v/v) Avicel PH-105 (DuPont, Wilmington, USA) in phosphate-buffered saline (PBS) was added and mixed. The mixture was centrifuged at 5,000×*g* for 1 min, upon which the supernatant was discarded. Proteins were extracted with PBS containing 0.5% Nonidet P-40 (Wako, Osaka, Japan) by sonication. The tissue suspension was incubated on ice for 1 h and centrifuged at 5000×*g* for 1 min at 4˚C. The obtained supernatants were used as tissue lysates.

## Lectin microarray

Lectin microarray analysis was performed as previously described [20, 22]. Twenty μL/tube of protein extracts was coupled with 10 μg of Cy3 succinimidyl ester (GE Healthcare, Buckinghamshire, UK) by incubation at room temperature for 1 h in the dark. After incubation, Probing Solution (GlycoTechnica Ltd, Yokohama, Japan) was added to make a total volume of 100 μL and incubated for 2 h at room temperature in the dark to inactivate any residual fluorescent reagent. Sixty μL of the Cy3-labeled protein solution diluted with Probing Solution was applied to the lectin microarray glass slide (LecChip™ Ver.1.0; GlycoTechnica Ltd; S1A Fig) and incubated for a minimum of 12 h at 20˚C in a humidified chamber with gentle shaking to allow for the complete formation of the lectin-glycan complex. Then, the lectin microarray was washed with the Probing Solution and scanned by GlycoStation™ Reader 1200 (GlycoTechnica Ltd). Image scanning was performed with the highest net intensity among 45 lectins to be around 50,000 arbitrary fluorescence units. Fluorescence intensities were analyzed using GlycoStation™ Signal Capture ver. 1.5 and GlycoStation™ ToolsPro Suite ver.1.5 (GlyoTechnica Ltd). The average intensity of the three spots for each lectin was normalized using the average value of 45 lectins. The normalized value was compared between the relapsed and the non-relapsed group. The origin and reported glycan specificities of the 45 lectins are listed in S1 Table.

## PNGase F treatment and sialidase treatment of relapsed TNBC cell lysates

For PNGase F treatment, 2 U of *N*-glycosidase F (PNGase F, Roche Diagnostics GmbH, Mannheim, Germany) was added to 15 μL of the Cy3-labeled protein solution and incubated for 8 h at 37˚C in the dark. For sialidase treatment, 10 mU of sialidase (Sialidase A, Prozyme Inc, California, US) and 1.5 μL of 0.3 M sodium acetate buffer (pH 5.2) were added to 15 μL of Cy3-labeled protein solution, and the mixture was incubated for 8 h at 37˚C in the dark. For both enzyme treatments, the treated solution was diluted with Probing Solution and applied to the lectin microarray and incubated for a minimum of 12 h at 20˚C in a humidified chamber with gentle shaking. After washing with Probing Solution, the lectin microarray was scanned

by GlycoStation Reader 1200 and images were obtained using the same exposure time and gain as used for analyzing untreated samples.

## TJA-II staining of TNBC surgical specimens

TJA-II (Medicago AB, Uppsala, Sweden) was labeled and purified using a Biotin Labeling Kit-NH2 (Dojindo, Kumamoto, Japan) according to the manufacturer's protocol. Biotinylated TJA-II staining was performed on 4-μm sections of formalin-fixed and paraffin-embedded surgical specimens. The sections used for TJA-II staining were consecutive with those used in the lectin microarray. Sections were deparaffinized and autoclaved with 10 mM citrate buffer (pH 6.0) for 10 min at 121°C. After cooling to room temperature and washing with distilled water, sections were transferred to an automated staining apparatus, BenchMark GX (Ventana Medical Systems, Oro Valley, AZ, USA). In BenchMark, tissues were incubated with 10 μg/mL of biotinylated TJA-II in PBS containing 3% bovine serum albumin (Calbiochem, La Jolla, CA) for 32 min at 37°C. Then, sections were stained with an iVIEW DAB Detection Kit (Ventana Medical Systems), and then counterstained with Hematoxylin Counterstain II (Ventana Medical Systems) and Bluing Reagent (Ventana Medical Systems) according to the manufacturer's protocol. Stained specimens were scanned by virtual slide system (VS-110, Olympus, Tokyo, Japan). The proportion of stained cells was estimated by dividing the number of stained tumor cells by the number of all tumor cells in a specified counting area by two trained pathologists using a microscope. The proportion score (PS) was defined as follows; PS 0: no staining, PS 1: <1/100, PS 2: 1/100-1/10, PS 3: 1/10-1/3, PS 4: 1/3-2/3, and PS 5: >2/3 cells stained [23].

## Ethical approval and informed consent

This study was carried out with approval from the Ethics Committee of Juntendo University Hospital (no. 20–276) and complies with the 1964 Helsinki Declaration and its later amendments or comparable ethical standards. No written or oral consent was obtained. All participants were informed that the research policy is available on the homepage of the hospital and that they have the opportunity to opt-out of the study at any time later on, which was approved by the Ethics Committee. The Ethics Committee approved of the opt-out method for the use of specimens and clinical data under the condition that all data are anonymized. In the present study, patient data were anonymized and patient samples were completely de-identified before being accessed for research purposes. None of the researchers had access to patient-identifiable information. Only those participants who had not opted-out from the study were included into the data analysis.

## Statistical analysis

Statistical analyses were performed using GraphPad Prism 6 software (GraphPad Software, San Diego, California USA, www.graphpad.com). Fisher's exact test (two-tailed)was used to evaluate differences in pathological size, nodal status, and histological type between the non-relapsed group and the relapsed group. Welch's t-test was used to evaluate differences in median age between the non-relapsed group and the relapsed group. Mann-Whitney U test was used to evaluate differences in median follow-up, Ki-67 Labeling Index (L.I.), lectin fluorescence intensity, and the PS of TJA-II staining between the non-relapsed group and the relapsed group. Paired t-test was used to evaluate differences in lectin fluorescence intensity before and after PNGase F or sialidase treatment of microdissected TNBC cells. A $p$-value smaller than 0.05 was considered statistically significant.

**Table 1. Patient characteristics and treatment modalities.**

| Clinicopathological variables | | Non-relapsed | Relapsed | *p*-value |
|---|---|---|---|---|
| Number | | 25 | 5 | |
| Median follow up (range) (months) | | 72 (47–123) | 42 (37–75) | 0.018[a] |
| Median age (range) (years) | | 58 (36–76) | 61 (52–71) | 0.27[b] |
| Pathological size (cm) | ≦2 | 10 | 1 | 0.63[c] |
| | > 2 | 15 | 4 | |
| Nodal status | negative | 20 | 3 | 0.57[c] |
| | positive | 5 | 2 | |
| Pathological stage | I | 10 | 0 | 0.22[d] |
| | II | 12 | 4 | |
| | III | 3 | 1 | |
| Histological type | IDC[e] | 21 | 4 | 1.0[c] |
| | Special type | 4 | 1 | |
| Nuclear grade | 1 and 2 | 8 | 1 | 0.67[d] |
| | 3 | 15 | 3 | |
| | unknown | 2 | 1 | |
| Ki-67 L.I. (median) | | 80 | 60 | 0.28[a] |
| Adjuvant therapy (n) | FEC + Taxane | 7 | 3 | 0.34[d] |
| | FEC only | 15 | 2 | |
| | Taxane only | 3 | 0 | |

[a]Mann-Whitney U test,

[b]Welch's t-test,

[c]Fisher's exact test,

[d]$\chi^2$ test.

[e]IDC: Invasive ductal carcinoma,

[f]FEC: 5-fluorouracil + epirubicin + cyclophosphamide.

## Results

### Patient characteristics

Thirty patients were included in the current study. The follow-up period was 37–75 months (median 42 months) in the relapsed group and 47–123 months (median 72 months) in the non-relapsed group. During the follow-up period, 5 patients relapsed (lung; $n = 2$, brain; $n = 1$, liver; $n = 1$, contralateral axillary lymph node; $n = 1$) and died. The median age was 58 and 61 years in the non-relapsed group and in the relapsed group, respectively. There were no differences in patient characteristics between the non-relapsed group and the relapsed group regarding pathological tumor size ($p = 0.63$), nodal status ($p = 0.57$), pathological stage ($p = 0.22$), histological type ($p = 1.0$), nuclear grade ($p = 0.67$) or Ki-67 L.I. ($p = 0.28$). In addition, there were no differences in the treatment modalities between the two groups ($p = 0.34$). Detailed patient characteristics and treatment modalities are presented in Table 1.

### TJA-II, ACA, WFA, and BPL bind more strongly to Cy3-labeled material in extracts of laser-microdissected TNBC cells from relapsed patients than from non-relapsed patients

TNBC cells resected from surgical specimens were lysed and extracts were prepared. Extracted proteins were labeled with Cy3 and subjected to lectin microarray analysis (Fig 2). The binding intensities of Cy3-labeled materials in the extracts of TNBC cells from 30 patient specimens

## Surgical specimens of TNBC patients

**Fig 2. Schematic overview of laser-assisted microdissection of surgical specimens and lectin microarray.**

with 45 lectins were quantified and compared (Fig 2, S1B Fig). The average fluorescence intensities of each lectin were compared between the relapsed and the non-relapsed group (Table 2). Seven lectins showed significant differences in binding intensity between TNBC groups (Fig 3). TJA-II ($p < 0.001$), *Amaranthus caudatus* agglutinin (ACA: $p = 0.013$), *Wisteria floribunda* agglutinin (WFA: $p = 0.018$), and *Bauhinia purpurea* lectin (BPL: $p = 0.026$) showed higher binding in the relapsed group than the non-relapsed group. The other three lectins, *Narcissus pseudonarcissus* agglutinin (NPA: $p = 0.027$), Concanavalin A (ConA: $p = 0.027$), and *Galanthus nivalis* agglutinin (GNA: $p = 0.031$) showed lower binding in the relapsed group than the non-relapsed group. The carbohydrate specificities of these lectins are listed in S1 Table. The results indicate that glycoproteins with glycans recognized by TJA-II, ACA, WFA, and BPL should potentially be useful as a marker and/or as a therapeutic target for the relapse-prone subset of TNBC.

Four μm-thick surgical specimens of TNBC patients were stained with hematoxylin, and a 2.5 mm$^2$ large area containing tumor cells was resected using laser-assisted microdissection. Soluble fractions were extracted with a NP40-based lysis buffer, and proteins were labelled with Cy3. The binding intensities of Cy3-labelled proteins to 45 lectins were quantified using lectin microarray, and the obtained glycan profiles were compared between the relapsed group and the non-relapsed group. TNBC; triple-negative breast cancer.

**Table 2. Comparison of the mean fluorescence intensity of 45 lectins between non-relapsed ($n = 25$) and relapsed ($n = 5$) triple-negative breast cancer (TNBC) patients.**

| Lectin | Non-relapsed group (Median) | Relapsed group (Median) | $p$-value[a] | Summary[b] |
|---|---|---|---|---|
| TJA-II | 36.93 | 105.84 | < 0.001 | Up |
| ACA | 36.71 | 85.09 | 0.013 | Up |
| WFA | 4.85 | 27.74 | 0.018 | Up |
| BPL | 5.29 | 30.34 | 0.026 | Up |
| NPA | 52.35 | 32.4 | 0.027 | Down |
| ConA | 209.61 | 165.52 | 0.027 | Down |
| GNA | 38.99 | 16.87 | 0.031 | Down |
| STL | 273.73 | 235 | 0.057 | - |
| AAL | 150.05 | 202.67 | 0.065 | - |
| MAH | 60.76 | 49.45 | 0.074 | - |
| SNA | 222.32 | 174.95 | 0.085 | - |
| PHA.L | 5.7 | 0 | 0.11 | - |
| AOL | 70.08 | 90.05 | 0.12 | - |
| ACG | 279.54 | 232.96 | 0.14 | - |
| UEA-I | 0.7 | 0 | 0.2 | - |
| Calsepa | 94.66 | 81.15 | 0.21 | - |
| TJA-I | 350.87 | 299.89 | 0.21 | - |
| ABA | 123.63 | 159.7 | 0.25 | - |
| MPA | 18.77 | 72.76 | 0.25 | - |
| TXLC-I | 46.05 | 57.41 | 0.28 | - |
| Jacalin | 153.4 | 195.6 | 0.3 | - |
| SSA | 257.37 | 195.37 | 0.3 | - |
| PHA.E | 76.21 | 54.65 | 0.3 | - |
| RCA120 | 181.4 | 192 | 0.36 | - |
| MAL-I | 25.1 | 16.01 | 0.36 | - |
| SBA | 7.083 | 35.92 | 0.39 | - |
| PSA | 50.56 | 46.26 | 0.42 | - |
| LEL | 381.62 | 377.76 | 0.52 | - |
| ECA | 3.11 | 2.79 | 0.61 | - |
| UDA | 503.26 | 473.62 | 0.71 | - |
| DBA | 5.98 | 0 | 0.71 | - |
| VVA | 7.16 | 11.23 | 0.72 | - |
| GSL-I-B4 | 0.63 | 2.74 | 0.78 | - |
| HPA | 0 | 0 | 0.81 | - |
| PTL-I | 0.78 | 0 | 0.83 | - |
| LTL | 7.05 | 3.71 | 0.84 | - |
| EEL | 0 | 0 | 0.84 | - |
| WGA | 208.54 | 203.02 | 0.87 | - |
| GSL.II | 0 | 0 | 0.88 | - |
| GSL-I-A4 | 6.11 | 4.99 | 0.91 | - |
| PWM | 1.61 | 1.82 | 0.91 | - |
| PNA | 0 | 0 | 0.92 | - |
| HHL | 31.06 | 25.09 | >0.99 | - |
| LCA | 59.92 | 62.19 | >0.99 | - |

(*Continued*)

**Table 2.** (Continued)

| Lectin | Non-relapsed group (Median) | Relapsed group (Median) | *p*-value[a] | Summary[b] |
|---|---|---|---|---|
| DSA | 283.86 | 294.5 | >0.99 | - |

TNBC cells were resected from sections of primary breast surgical specimens using laser-assisted microdissection and subjected to microarray analysis.

[a]Mann-Whitney U test was used to compare fluorescence intensities of each lectin between the relapsed and the non-relapsed group.

[b]Describes how the lectin binding of the relapsed group changed compared to the non-relapsed group.

### TJA-II and ACA bind *O*-glycans of glycoproteins from TNBC cells

To obtain information on the nature of the binding sites for lectins that showed significant differences in the binding between the relapsed and the non-relapsed group, Cy3-labeled TNBC tissue lysates were treated with PNGase F or sialidase. PNGase F catalyzes the cleavage of whole *N*-glycans from glycoproteins, while sialidase catalyzes the cleavage of terminal sialic acid from *N*- and *O*-glycans. After PNGase F treatment, the binding to TJA-II and ACA did not substantially change (Fig 4A and 4B). In contrast, the binding to WFA, BPL, and ConA decreased, and the binding to NPA and GNA disappeared almost completely (except the binding of the extract from patient No. 8 to NPA) (Fig 4C–4G). These results indicate that TJA-II and ACA mainly bound *O*-glycans, whereas NPA and GNA bound *N*-glycans. The binding sites for WFA, BPL, and ConA apparently consisted of *N*- and *O*-linked glycans. The bindings of TJA-II, WFA, BPL, and NPA increased after sialidase treatment (Fig 4H, 4J, 4K and 4L). Such increased binding was not observed with ACA, ConA, or GNA (Fig 4I, 4M and 4N).

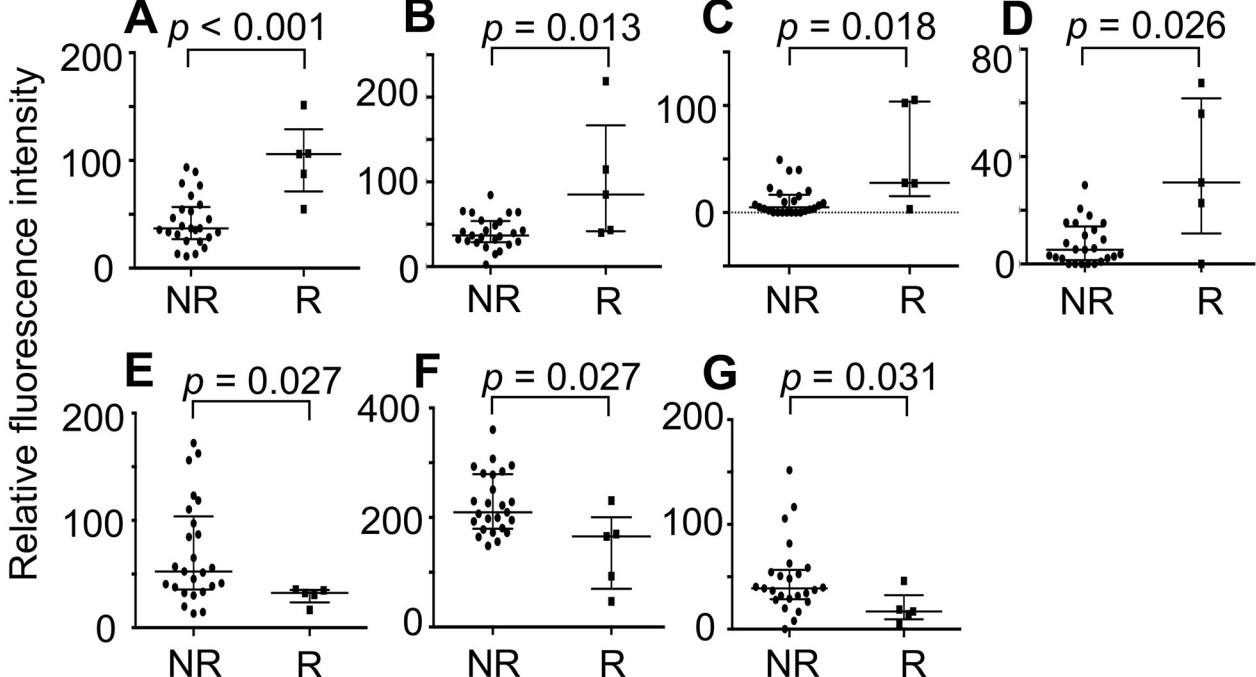

**Fig 3. Relative fluorescence intensities of 7 lectins that showed significantly different binding between non-relapsed (*n* = 25) and relapsed (*n* = 5) TNBC patients.** (**A**) TJA-II, (**B**) ACA, (**C**) WFA, (**D**) BPL, (**E**) NPA, (**F**) ConA, (**G**) GNA. Median ± quartiles are shown. Mann-Whitney U test was used for statistical analysis. ACA; *Amaranthus caudatus* agglutinin, BPL; *Bauhinia purpurea* lectin, ConA; Concanavalin A, GNA; *Galanthus nivalis* agglutinin, NPA; *Narcissus pseudonarcissus* agglutinin, NR; Non-relapsed, R; Relapsed, SEM; standard error of the mean, TJA-II; *Trichosanthes japonica* agglutinin-II, TNBC; triple-negative breast cancer, WFA; *Wisteria floribunda* agglutinin.

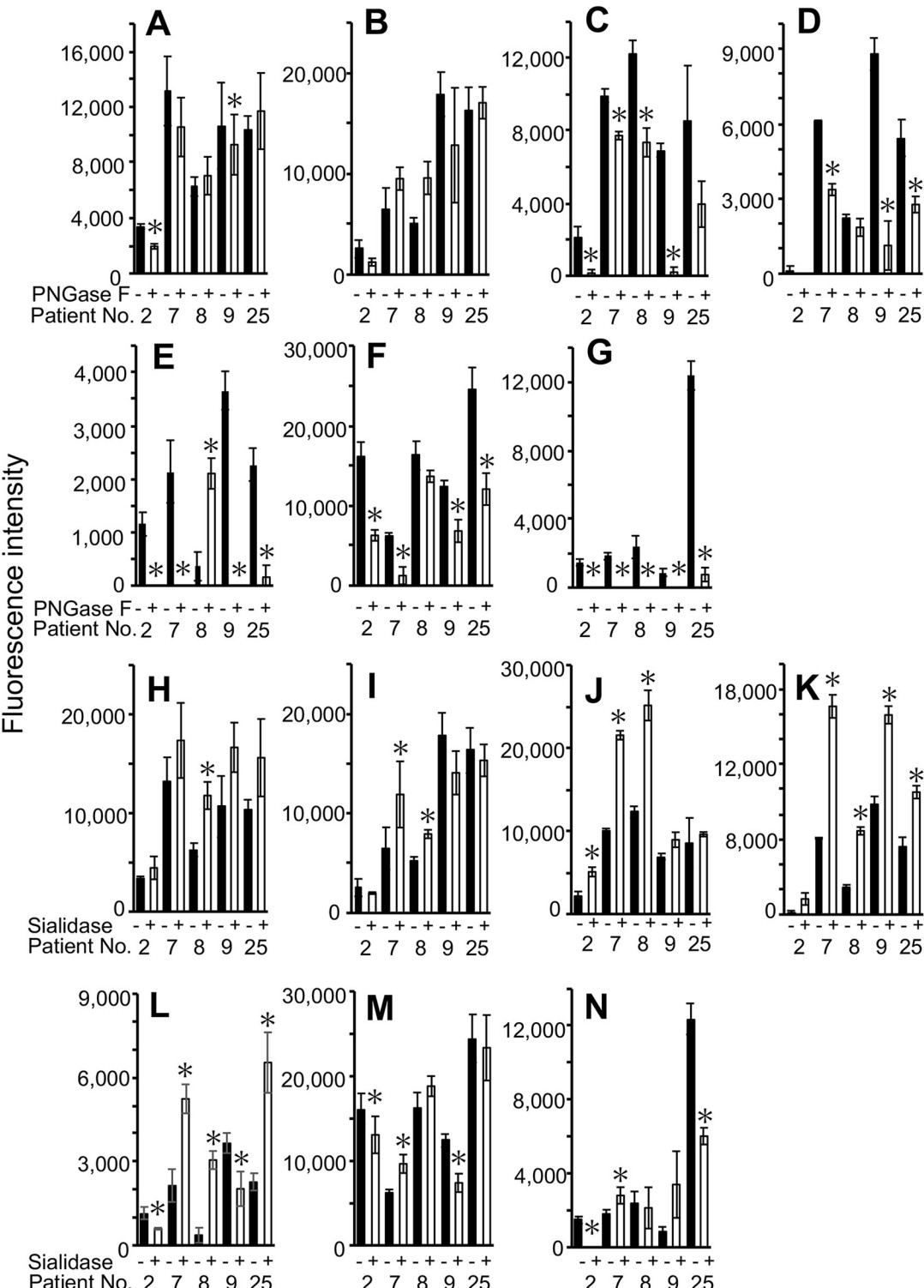

**Fig 4. Changes in lectin binding intensities after PNGase F and sialidase treatment of relapsed TNBC cell lysates.** (**A-G**) Changes in lectin binding intensity after PNGase F treatment. (**A**) TJA-II, (**B**) ACA, (**C**) WFA, (**D**) BPL, (**E**) NPA, (**F**) ConA, (**G**) GNA. (**H-N**) Changes in lectin binding intensity after sialidase treatment. (**H**) TJA-II, (**I**) ACA, (**J**) WFA, (**K**) BPL, (**L**) NPA, (**M**) ConA, (**N**) GNA. Black bars: no treatment, white bars: enzyme treatment. ACA; *Amaranthus caudatus* agglutinin, BPL; *Bauhinia purpurea* lectin, ConA; Concanavalin A, GNA; *Galanthus nivalis* agglutinin, NPA; *Narcissus pseudonarcissus* agglutinin, TJA-II; *Trichosanthes japonica* agglutinin-II, TNBC; triple-negative breast cancer, WFA; *Wisteria floribunda* agglutinin. $^*p < 0.05$ (paired t-test, treatment versus no treatment).

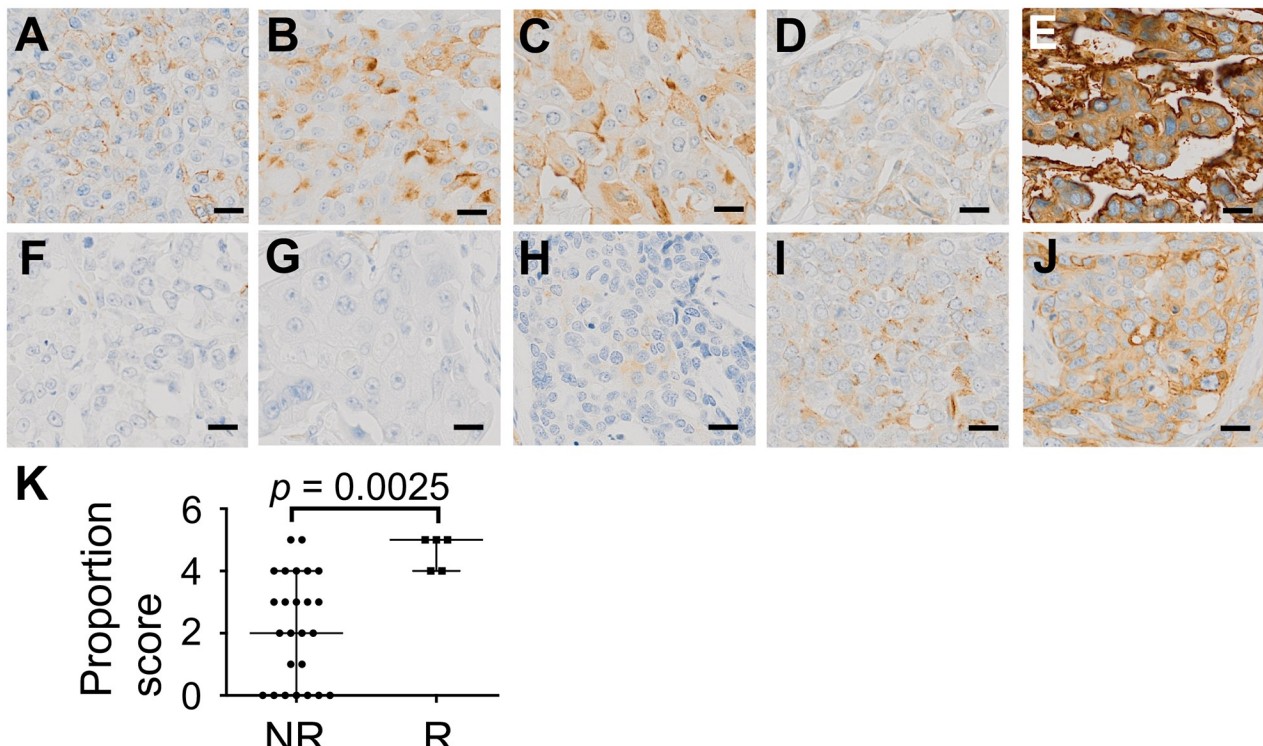

**Fig 5. Representative images of TNBC tissues stained with TJA-II lectin.** (**A-E**) Results of relapsed patients. (**A**) No. 2; PS 4, (**B**) No. 25; PS 4, (**C**) No. 7; PS 5, (**D**) No. 8; PS 5, (**E**) No. 9; PS 5. (**F-J**) Results of non-relapsed patients. (**F**) No. 22; PS 0, (**G**) No. 35; PS 0, (**H**) No. 11; PS 2, (**I**) No. 28; PS 3, (**J**) No.1 0; PS 4. Scale bars show 20 μm. (**K**) The PS of TJA-II staining in the non-relapsed group ($n = 25$) and the relapsed group ($n = 5$). Median ± quartiles are shown. Mann-Whitney U test was used for statistical analysis. The PS was estimated by two pathologists. NR; Non-relapsed, R; Relapsed, PS; Proportion score, TJA-II; *Trichosanthes japonica* agglutinin-II, TNBC; triple-negative breast cancer.

## TJA-II stained proportionally more TNBC cells in tumor sections from relapsed patients than non-relapsed patients

We decided to further pursue TJA-II because the presence of higher levels of β-linked terminal *N*-acetylgalactosaminyl (GalNAc) residues in tumor cells of relapsed patients was supported by the binding of other lectins with similar specificity. We used TJA-II as a histochemical tool and assessed whether elevated TJA-II binding could be observed in sections from relapsed tumor specimens. Fig 5A–5J shows representative images of TNBC tissues stained with TJA-II. TJA-II binding was observed in some but not all tissues, and the staining profiles were heterogeneous. It was notable that strong binding was observed with cancerous cells but not with other types of cells in the tumor region, such as stromal cells or invading cells of the immune system. The staining of TNBC cells was heterogeneous with regard to the cell-to-cell differences and subcellular localizations. Staining was particularly strong with cell surface membranes and in the cytoplasm of TNBC cells.

The proportion of cells stained by TJA-II was evaluated under the microscope and expressed as PS. Relapsed patients had a PS between 4 and 5 (Fig 5A–5E). In contrast, the PS of non-relapsed patients ranged from 0 to 5 (Fig 5F–5J). Taken together, the PS of TJA-II staining was significantly higher in the relapsed group than the non-relapsed group ($p = 0.0025$) (Fig 5K), in line with the lectin microarray results. S2 Fig shows the relation between the proportion score of TJA-II staining, clinicopathological variables, and adjuvant

therapy in relapsed and the non-relapsed TNBC patients. For all variables shown, the PS tends to be higher in the relapsed group than the non-relapsed group.

## Discussion

In the present study, we obtained TNBC cells from 30 surgical specimens using laser microdissection and searched for glycans attached to proteins characteristic to relapse-prone patients using lectin microarrays. Among 45 lectins, the glycoprotein binding to four lectins TJA-II, ACA, WFA, and BPL was higher in relapsed patients than in non-relapsed patients (Fig 3A–3D). Glycoproteins recognized by TJA-II are known to contain glycans having fucosyl (Fuc) α1–2 Gal β1 sequences or β-linked terminal GalNAc residues (S1 Table) [24]. However, no difference in the binding of *Ulex europaeus* agglutinin I (UEA-I) or *Lotus tetragolonobus* lectin (LTL), known to have affinity with Fuc α1–2 Gal β1 (S1 Table), was observed between the relapsed and the non-relapsed group (Table 2). As for higher levels of β-linked terminal GalNAc residues in TNBC cells from relapsed patients than non-relapsed patients, this possibility is supported by the present results showing that the binding intensity of WFA and BPL, which also recognize β-linked terminal GalNAc residues (S1 Table), was elevated in the relapsed group (Fig 3C and 3D). PNGase F treatment of TNBC cell lysates from relapsed patients did not considerably alter TJA-II reactivity, suggesting that the GalNAc residues recognized by TJA-II mainly reside on *O*-glycans (Fig 4A). After the same treatment, the reactivity with WFA and BPL was partially reduced, indicating that these lectins bound both *N*- and *O*-glycans (Fig 4C and 4D). Taken these results together, it can be reasonably speculated that glycoproteins bearing *O*-glycans with β-linked terminal GalNAc residues, recognized by TJA-II, WFA, and BPL, are present at high levels in TNBC cells from relapsed patients. WFA and BPL are likely to bind also *N*-glycans on the same or other glycoproteins. Whether glycoproteins from relapsed TNBC patients reactive with TJA-II also bind ACA is currently unknown. Although T-antigen (terminal Gal β1-GalNAc) and sialylated T-antigen (S1 Table) are the primary binding structures for ACA, lectins with similar specificity, such as Jacalin, *Maclura pomifera* agglutinin (MPA), and *Arachis hypogaea* agglutinin (PNA), did not show elevated reactivity with extracts from TNBC cells of relapsed patients (Table 2).

There are 6 glycosyltransferases which are potentially capable of transferring GalNAc to glycans, namely β-1,3-*N*-acetyl-galactosaminyltransferase (B3GALNT) 1 and 2, and β-1,4-*N*-acetyl-galactosaminyltransferase (B4GALNT) 1, 2, 3, and 4. Which of these enzymes is responsible for transferring GalNAc residues (β linkage) to *O*-glycans is unknown. We searched for previous papers that investigated whether the expression of one or more of these six genes correlates with survival in TNBC patients. Matsuo and co-workers reported that *B3GALNT2* was expressed at high levels in TNBC cell lines and that downregulation of *B3GALNT2* by siRNA resulted in a significant reduction of cell growth through the induction of apoptosis [25]. However, they did not investigate whether manipulating *B3GALNT2* gene levels led to any changes in glycan structures. Thus, whether the levels of terminal GalNAc residues determine the TNBC cell growth rate is currently unknown. It is also widely accepted that expression of glycosyltransferases is involved in multidrug resistance in cancer, though the mechanism still remains elusive [26]. Another important issue is that the addition of terminal GalNAc residues to glycoproteins may alter the interactions of this glycoprotein with endogenous lectins such as galectins [27]. Galectins form a lattice on cell surfaces with glycoproteins containing its ligands. Such rearrangement is known to modify the function of drug transporters, resulting in the modification of drug sensitivity [28]. Further investigations should be conducted to reveal the mechanism by which TNBC patients with tumor cells expressing TJA-II-reactive

glycans tend to relapse early. To this end, the most important task is to identify the TNBC cellular glycoprotein(s) to which TJA-II binds.

The uniqueness of our present study lies in the fact that all investigations were conducted on surgical specimens and the findings have a direct clinical relevance. Histochemical staining of tissues with TJA-II is potentially useful to predict patients who are likely to experience early relapse. However, to understand the biological mechanism by which TJA-II-reactive TNBC cells behave differently from other TNBC cells, studies with experimental models should also be conducted, which we are currently pursuing.

In the present study, although the number of analyzed cases was small, a significant difference was found between the non-relapsed and the relapsed group. However, in the non-relapsed group, 10 patients had a follow-up period of less than 5 years and still require continued observation.

In conclusion, we found that cell extracts from surgical specimens of relapsed TNBC patients had a unique lectin reactivity, showing higher levels of TJA-II (also WFA and BPL) binding than in non-relapsed patients. Histochemical TJA-II staining of adjacent sections confirmed this difference. Identifying the glycans having β-linked terminal GalNAc residues to which TJA-II binds and their carrier glycoprotein(s) using glycomics/glycoproteomics technologies should lead to the discovery of new target molecules for the effective therapy of the relapse-prone TNBC subset.

## Supporting information

**S1 Fig. Lectin microarray of 30 TNBC patients. (A)** Layout of 45 lectins on the LecChip™ Ver.1.0. **(B)** Lectin binding patterns for all 30 cases. The lectin number is the number listed before the lectin name in (A). Patient numbers written in red belong to the relapsed group, and patient numbers written in black belong to the non-relapsed group.
(TIF)

**S2 Fig. Relation between the Proportion Score (PS) of TJA-II staining, clinicopathological variables, and adjuvant therapy in non-relapsed and relapsed TNBC patients.** Surgical specimens from TNBC patients were stained with TJA-II lectin. Lectin staining intensity was evaluated by two pathologists and expressed as PS as explained in the Material and method section. Each dot represents the PS of one individual patient. **(A)** Pathological size, **(B)** Nodal status, **(C)** Pathological stage, **(D)** Histological type, **(E)** Nuclear grade, **(F)** Adjuvant therapy. For each variable A-F, non-relapsed and relapsed patients are shown divided into subcategories. Median ± quartiles are shown. Mann-Whitney U test. FEC; 5-fluorouracil + epirubicin + cyclophosphamide, IDC; Invasive ductal carcinoma, NR; Non-relapsed, R; Relapsed, T; Taxane.
(TIF)

**S1 Table. Lectin abbreviations, lectin origins, and reported glycan specificities of 45 lectins on the LecChip™ Ver.1.0.**
(DOCX)

## Acknowledgments

We wish to thank Dr. Yasunori Chiba (Department of Life Science and Biotechnology, Biotechnology Research Institute for Drug Discovery) for valuable discussions and Dr. Kouji Yamamoto (Department of Biostatistics, Yokohama City University School of Medicine) and Dr. Naotake Yanagisawa (Juntendo Clinical Research and Trial Center) for statistical advice. We would like to thank the members of the Department of Human Pathology, Juntendo

University School of Medicine, for preparing surgical specimens. We also wish to thank the members of the Laboratory of Morphology and Image Analysis, Research Support Center, Juntendo University Graduate School of Medicine, for technical assistance.

## Author Contributions

**Conceptualization:** Kaori Denda-Nagai, Mitsue Saito, Tatsuro Irimura.

**Data curation:** Madoka Sakata-Matsuzawa, Haruhiko Fujihira, Miki Noji.

**Formal analysis:** Madoka Sakata-Matsuzawa.

**Investigation:** Madoka Sakata-Matsuzawa.

**Methodology:** Atsushi Matsuda, Atsushi Kuno, Katsuya Nakai, Yoshiya Horimoto.

**Project administration:** Kaori Denda-Nagai.

**Resources:** Misato Okazaki, Katsuya Nakai, Yoshiya Horimoto.

**Supervision:** Kaori Denda-Nagai, Mitsue Saito, Tatsuro Irimura.

**Writing – original draft:** Madoka Sakata-Matsuzawa, Haruhiko Fujihira.

**Writing – review & editing:** Kaori Denda-Nagai, Katrin Beate Ishii-Schrade, Tatsuro Irimura.

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
