## [Decision Letter · Decision Letter 0]

13 Apr 2021

Glycans unique to the relapse-prone subset within triple-negative breast cancer as revealed by lectin array-based analysis of surgical specimens

PONE-D-21-03731

Dear Professor Irimura,

We’re pleased to inform you that your manuscript has been judged scientifically suitable for publication and will be formally accepted for publication once it meets all outstanding technical requirements.

Kind regards,

Lu-Gang Yu, PhD

Academic Editor

PLOS ONE

Additional Editor Comments:

The study was well designed and well executed, data analysis was careful and the conclusion was supported by the results.

Journal Requirements:

1. We note that you are reporting an analysis of a microarray, next-generation sequencing, or deep sequencing data set. PLOS requires that authors comply with field-specific standards for preparation, recording, and deposition of data in repositories appropriate to their field. Please upload these data to a stable, public repository (such as ArrayExpress, Gene Expression Omnibus (GEO), DNA Data Bank of Japan (DDBJ), NCBI GenBank, NCBI Sequence Read Archive, or EMBL Nucleotide Sequence Database (ENA)). In your revised cover letter, please provide the relevant accession numbers that may be used to access these data. For a full list of recommended repositories, see http://journals.plos.org/plosone/s/data-availability#loc-omics or http://journals.plos.org/plosone/s/data-availability#loc-sequencing

2. Thank you for providing the date(s) when patient medical information was initially recorded. Please also include the date(s) on which your research team accessed the (i) databases and (ii) samples used in your study.

3. For the  66 TNBC patients initially screened, please provide additional details regarding participant consent. In the ethics statement in the Methods and online submission information, please ensure that you have specified:

 - whether consent was obtained

 - whether consent was informed

 - what type of consent you obtained (for instance, written or verbal, and if verbal, how it was documented and witnessed)

 - if consent was not obtained, state if researchers had access to any personal identifying information of the 66 patients, and whether the data was anonymized before researchers accessed it."

4. In your ethics statement in the online submission form and in your manuscript text, for the immunohistochemistry specimens please clarify:

 - whether the ethics committee specifically approved the opt-out consent method described for the use of the immunohistochemistry specimens

 - whether this consent was informed

 - whether researchers had access to any personal identifying information of the 30 patients who donated samples, and whether samples were completely de-identified before researchers accessed them

Reviewers' comments:

Reviewer's Responses to Questions

**Comments to the Author**

1. Is the manuscript technically sound, and do the data support the conclusions?

Reviewer #1: Yes

2. Has the statistical analysis been performed appropriately and rigorously? 

Reviewer #1: Yes

3. Have the authors made all data underlying the findings in their manuscript fully available?

Reviewer #1: Yes

4. Is the manuscript presented in an intelligible fashion and written in standard English?

Reviewer #1: Yes

5. Review Comments to the Author

Reviewer #1: The paper was well designed and written.

Results are interesting and were presented appropriately supporting the conclusions.

All findings were supported by available data and by statistical analysis.

Tables and figures are self-explaining and figures' legends are sufficient for isolated understanding.

Discussion was rich and well written.

6. PLOS authors have the option to publish the peer review history of their article (what does this mean?). If published, this will include your full peer review and any attached files.

Reviewer #1: No

---

## [Editor Report · Acceptance letter]

30 Apr 2021

PONE-D-21-03731 

Glycans unique to the relapse-prone subset within triple-negative breast cancer as revealed by lectin array-based analysis of surgical specimens 

Dear Dr. Irimura:

I'm pleased to inform you that your manuscript has been deemed suitable for publication in PLOS ONE. Congratulations! Your manuscript is now with our production department. 

Kind regards, 

on behalf of

Professor Lu-Gang Yu 

Academic Editor

PLOS ONE